# A Study of Biomarkers Associated with Metabolic Dysfunction-Associated Fatty Liver Disease in Patients with Type 2 Diabetes

**DOI:** 10.3390/diagnostics12102426

**Published:** 2022-10-07

**Authors:** Ion Cristian Efrem, Maria Moța, Ionela Mihaela Vladu, Adina Mitrea, Diana Clenciu, Diana Cristina Protasiewicz Timofticiuc, Ileana-Diana Diaconu, Adina Turcu, Anda Elena Crișan, Cristiana Geormăneanu, Adina Dorina Glodeanu, Beatrice Mahler, Marinela Sînziana Tudor, Anca Maria Amzolini, Simona Elena Micu, Anca Barău Abu Alhija, Adrian Mită, Maria Monalisa Filip, Maria Forțofoiu

**Affiliations:** 1Department of Internal Medicine and Medical Semiology, Faculty of Dentistry, University of Medicine and Pharmacy of Craiova, 200349 Craiova, Romania; 2Department of Internal Medicine with Gastroenterology Compartment, “Philanthropy” Clinical Municipal Hospital of Craiova, 200143 Craiova, Romania; 3Doctoral School, University of Medicine and Pharmacy of Craiova, 200349 Craiova, Romania; 4Department of Diabetes, Nutrition and Metabolic Diseases, Faculty of Medicine, University of Medicine and Pharmacy of Craiova, 200349 Craiova, Romania; 5Department of Diabetes, Nutrition and Metabolic Diseases, County Clinical Emergency Hospital of Craiova, 200642 Craiova, Romania; 6Department of Diabetes, Nutrition and Metabolic Diseases, “Philanthropy” Clinical Municipal Hospital of Craiova, 200143 Craiova, Romania; 7Department of Pediatric Pneumology, “Marius Nasta” National Institute of Pneumology of Bucharest, 050159 Bucharest, Romania; 8Department of Oro-Dental Prevention and Oral Health, Faculty of Dentistry, University of Medicine and Pharmacy of Craiova, 200349 Craiova, Romania; 9“Victor Babes” Infectious Diseases Hospital, 200515 Craiova, Romania; 10Department of Oncology, Faculty of Medicine, University of Medicine and Pharmacy of Craiova, 200349 Craiova, Romania; 11Department of Radiotherapy, County Clinical Emergency Hospital of Craiova, 200642 Craiova, Romania; 12Department of Emergency Medicine, Faculty of Medicine, University of Medicine and Pharmacy of Craiova, 200349 Craiova, Romania; 13Department of Emergency Medicine, County Clinical Emergency Hospital of Craiova, 200642 Craiova, Romania; 14Department of Cardiology, “Philanthropy” Clinical Municipal Hospital of Craiova, 200143 Craiova, Romania; 15Department of Pneumology, University of Medicine and Pharmacy “Carol Davila”, 020021 Bucharest, Romania; 16Institute of Pneumophtisiology “Marius Nasta”, 050159 Bucharest, Romania; 17Department of Internal Medicine, Strehaia External Section of Drobeta Turnu-Severin Emergency County Hospital, 225300, Strehaia, Romania; 18Department of Medical Semiology, Faculty of Medicine, University of Medicine and Pharmacy of Craiova, 200349 Craiova, Romania; 19Department of Medical Semiology, Faculty of Midwife and Nursing, University of Medicine and Pharmacy of Craiova, 200349 Craiova, Romania; 20Department of Emergency Medicine, “Philanthropy” Clinical Municipal Hospital of Craiova, 200143 Craiova, Romania

**Keywords:** non-alcoholic fatty liver disease, type 2 diabetes, insulin resistance, metabolic dysfunction-associated fatty liver disease, insulin resistance, biomarkers, predictors, diagnosis

## Abstract

Metabolic dysfunction-associated fatty liver disease (MAFLD) is a new term that no longer excludes patients that consume alcohol or present other liver diseases, unlike nonalcoholic fatty liver disease (NAFLD). The aim of this study was to evaluate the role of different biomarkers as predictors of MAFLD in patients with type 2 diabetes mellitus (T2DM). In this regard, a cross-sectional, non-interventional study was conducted over a period of 8 months in patients with T2DM. Liver steatosis displayed by abdominal ultrasound certified the MAFLD diagnosis. A percentage of 49.5% of the studied patients presented MAFLD. Through logistic regression adjusted for gender, age, T2DM duration, lipid-lowering therapy, smoking status, nutritional status, we demonstrated that elevated triglycerides (TG) levels, high non-high-density-lipoprotein (HDL)-cholesterol-to-HDL-cholesterol (non-HDL/HDL) ratio, high atherogenic index of plasma (AIP), and increased Homeostatic Model Assessment for Insulin Resistance (HOMA-IR) had predictive value for MAFLD in patients with T2DM. Furthermore, we calculated the optimal cut-off values for these biomarkers (184 mg/dL for TG, 0.615 for AIP, 3.9 for the non-HDL/HDL ratio, and 2.01 for HOMA-IR) which can predict the presence of MAFLD in patients with T2DM. To our knowledge, this is the first study to assess the predictive value of the non-HDL/HDL ratio for MAFLD in patients with T2DM.

## 1. Introduction

For many years, fatty liver disease was artificially classified according to the patient’s alcohol consumption into two categories: alcoholic liver disease (ALD) and non-alcoholic fatty liver disease (NAFLD) [1], the latter being defined for the first time over 35 years ago [2,3]. This term covers diverse liver lesions, ranging from steatosis to cirrhosis, in the absence of significant alcohol consumption [4,5]. NAFLD is the most frequent liver disease, affecting up to 25% of adults worldwide, the highest prevalence being registered in South America and the Middle East, while in Africa registered the lowest prevalence [6,7]. Moreover, this disease has an increasing prevalence, as studies estimate that over 100 million people will be affected by NAFLD in the United States by 2030 [8]. What is even more worrisome is the fact that an important number of these people will present non-alcoholic steatohepatitis (NASH), which is the progressive form of NAFLD. In this regard, it is understandable that the high risk of progression to liver cirrhosis and even to hepatocellular carcinoma, as well as outcomes such as cardiovascular diseases, make NAFLD an important public health concern [6,7,9].

The gold standard for the diagnosis of NAFLD is liver biopsy. However, due to the adverse effects that may be associated with this procedure (pain, bleeding, and in rare cases, even death) NAFLD is diagnosed in the clinical practice by the measurement of increased levels of liver enzymes and abdominal ultrasound that evidences liver steatosis [10,11].

Studies have demonstrated an association between NAFLD and metabolic disorders (obesity, insulin resistance, type 2 diabetes mellitus (T2DM)), cardiovascular diseases, and gut microbiome disruptions [4,12].

In this regard, a meta-analysis evidenced that NAFLD (diagnosed on the basis of abnormal levels of liver enzymes and abdominal ultrasound) was associated with a twofold increased risk of T2DM development as well as metabolic syndrome development over a 5-year median follow-up [11]. Furthermore, Younossi et al., in a meta-analysis including over 8.5 million people from 22 countries, reported high frequencies of metabolic diseases in patients with NAFLD, corresponding to 22.51% for T2DM, 39.34% for arterial hypertension, 42.54% for metabolic syndrome, 51.34% for obesity, and 69.16% for dyslipidemia [13]. Furthermore, the same study recorded a high prevalence of NASH (59.1%) in patients with NAFLD who underwent a liver biopsy [13]. Interestingly, the patients with NASH also presented a higher prevalence of metabolic disorders, e.g., 43.63% for T2DM, 72.13% for dyslipidemia, and 81.83% for obesity [13].

This high frequency of metabolic disorders associated with NAFLD led to the newly defined term “metabolic dysfunction-associated fatty liver disease” (MAFLD) [14]. It is important to mention that the two terms, NAFLD and MAFLD, do not completely overlap, as the diagnosis of MAFLD requires the evidence of liver steatosis defined based on a liver biopsy, biomarkers, or imaging associated with metabolic dysregulations, T2DM, and overweight or obesity [15,16]. Another reason for this nomenclature change is related to the bias linked to the evaluation of alcohol consumption, patients with mild to moderate alcohol consumption being diagnosed with NAFLD, while subjects with metabolic disorders and high alcohol intake being diagnosed with ALD [1]. In the clinical practice, it is challenging to classify patients in either category, as in many cases there is an association between mild to moderate alcohol consumption and metabolic impairments [1,17,18]. In addition, in a study recently conducted by Lin et al., it was proved that patients with MAFLD have a higher proportion of abnormal fibrosis scores when compared to patients diagnosed with NAFLD, leading the authors to conclude that using the new terminology is more appropriate, as this concept better identifies patients at a high risk of liver disease progression [1]. Furthermore, it was demonstrated that patients with T2DM diagnosed with NAFLD/MAFLD are also at a higher risk for both micro- and macrovascular chronic complications linked to diabetes [19,20,21]. Additionally, patients with MAFLD are at risk of developing hepatic complications, such as cirrhosis and hepatocellular carcinoma, sometimes requiring a liver transplant [22]. Taking into account all these aspects, together with the increase in prevalence of both MAFLD and T2DM, it is understandable that both diseases are major public health issues, with important societal and economic burdens. Therefore, it is of utmost importance to identify patients with MAFLD that are at risk of developing complications [11,23]. It is also important to identify non-invasive procedures and biomarkers that can not only diagnose MAFLD, but also stratify the disease according to the risk of developing complications, in order to select the patients who require a pharmacological treatment [23].

An early diagnosis of MAFLD in patients with T2DM may lead to an important decrease of morbidity and mortality in these patients [24]. The atherogenic index of plasma (AIP), defined as the logarithm of the triglyceride- (TG) to-high-density lipoprotein-cholesterol (HDL-C) ratio [25], was proposed as a non-invasive screening tool for MAFLD [24], as previous studies have evidenced the relationship between this parameter and NAFLD [25,26,27].

The aim of this study was to assess the usefulness of different blood biomarkers in patients with MAFLD and T2DM.

## 2. Materials and Methods

A cross-sectional, non-interventional study was conducted over a period of 8 months (January to August 2022). The study enrolled 200 subjects with T2DM (Figure 1) who were screened for MAFLD using abdominal ultrasound, which resulted in dividing the study subjects into 2 groups:-Group 1: 101 subjects without MAFLD-Group 2: 99 subjects with MAFLD

The study inclusion criteria used in the study were: adult patients (age ≥18 years) diagnosed with T2DM treated with oral and/or injectable non-insulin antidiabetic agents, recruited from the Outpatient Department of Diabetes, Nutrition and Metabolic Diseases of the Municipal Clinical Hospital “Philanthropy” of Craiova, after reading and signing the study informed consent. The exclusion criteria were: subjects under the age of 18 years, patients diagnosed with type 1 diabetes mellitus and other types of diabetes, patients with T2DM treated with insulin, severe uncontrolled psychiatric pathology, severe comorbidities (heart disease, renal diseases, cancer), patients who underwent bariatric surgery, as well as patients with other conditions that could influence the results of the investigations.

All the patients enrolled in the study participated voluntarily and were enrolled only after the informed consent form was read and signed. We conducted the study according to the guidelines of the Declaration of Helsinki, respecting the right to integrity and confidentiality, in accordance with good clinical practice. The patients were given the option to withdraw from the study at any time. The study was performed with the approval of the Ethics Committee of the University of Medicine and Pharmacy of Craiova, Romania (162/19.08.2022).

For all the patients, we recorded demographic data (age and gender), anthropometric measurements (weight, height, waist circumference (WC), hip circumference (HC)), as well as any relevant data from a physical examination. The body mass index (BMI) was calculated using the formula: BMI = weight (in kilograms)/height^2^ (in meters), and the nutritional status of the subjects was classified according to World Health Organization criteria [28]. The measurement of the WC was performed at the midpoint between the lower border of the rib cage and the upper iliac crest, while the HC was measured over the femoral trochanters. The waist-to-hip ratio (WHR) and the waist-to-height ratio (WHtR) were also calculated.

Venous blood samples were collected from the study patients, and the following blood tests were performed: fasting plasma blood glucose (FPG), fasting insulinemia, glycated hemoglobin (HbA1c), aspartate aminotransferase (AST), alanine aminotransferase (ALT), lipid profile (total cholesterol, HDL-cholesterol, low-density lipoprotein (LDL)-cholesterol, TG). AIP, non-HDL-cholesterol, TG/HDL ratio, non-HDL/HDL ratio, and ALT/AST ratio were also calculated. Lipid disorders were defined based on an abnormal lipid profile and/or a lipid lowering treatment.

Insulin resistance, a condition associated with MAFLD, was assessed by the Homeostatic Model Assessment for Insulin Resistance (HOMA-IR) calculated using the formula: (FPG (mg/dL) × fasting insulinemia (mU/L))/405. The visceral adiposity index (VAI), another parameter associated with insulin resistance, was calculated using the following formulas [29]: VAI = [WC/39.68 + (1.88 × BMI)] × (TG/1.03) × (1.31/HDL) for males and VAI = [WC/36.58 + (1.89 × BMI)] × (TG/0.81) × (1.52/HDL) for females.

MAFLD was diagnosed by the presence of liver steatosis on abdominal ultrasound. This examination was performed by the same 3 physicians in all the studied subjects.

All the data were recorded in an EXCEL database and afterwards analyzed utilizing the Statistical Package for the Social Sciences (SPSS) version 26.0 (SPSS Inc., Chicago, IL, USA). Using the Kolmogorov–Smirnov test, we assessed the normality of continuous variables, data with a normal distribution being reported as mean ± standard deviation (SD), while data with an abnormal distribution being reported as median and interquartile range (IQR).

The statistical significance of the differences between groups was evaluated using the Mann–Whitney U test to compare the medians (for variables with an abnormal distribution) and the t-test and ANOVA test to compare the means (for variables with a normal distribution). The area under the receiving operating curve and the logistic regression were used to assess the associations between the studied parameters and MAFLD. The statistical significance threshold was defined as *p* < 0.05.

## 3. Results

In this study, we found a frequency of MAFLD of 49.5% in patients with T2DM with a median diabetes duration of 6 years. The mean age of the patients enrolled in the study was 56.55 ± 9.32 years, and the patients were almost equally divided according to gender (51.55% females). Given the association between obesity and MAFLD, we studied differences regarding anthropometric parameters in the two study groups. As it can be observed in Table 1, there were statistically significant differences regarding BMI, WC, and WHtR between the subjects with MAFLD and the ones without MAFLD only for female patients, while for WHR, we did not find statistically significant differences for both genders.

Regarding the biological parameters (Table 2), we observed that, with the exception of AST, all the examined biomarkers presented statistically significant higher median values in the patients with T2DM and MAFLD. Furthermore, in this study, 89% of the patients presented lipid disorders.

Taking into account that the smoking status can influence lipid parameters, we intended to analyze the relationships between MAFLD and smoking. In our study population, the frequency of actively smoking patients was 18.5% (37 patients), while 23 subjects declared to be former smokers (11.5%). As it is shown in Table 3, only TG, TG/HDL ratio, AIP, and VAI presented significantly higher medians in active smokers.

Through backward stepwise regression, adjusting for gender, age, T2DM duration, lipid-lowering therapy, smoking status, BMI, and WC, we demonstrated that only TG, non-HDL/HDL ratio, AIP, and HOMA-IR were independent predictors for MAFLD in patients with T2DM (Table 4).

Furthermore, we also analyzed the AUROC for the biomarkers independently associated with MAFLD and determined the cut-off points suggestive of MAFLD in patients with T2DM for the four independent predictors (Figure 2 and Table 5).

## 4. Discussion

Currently, NAFLD is the most frequent cause of chronic liver disease worldwide, being an important risk factor for the development of liver cirrhosis and carcinoma [30,31,32]. Moreover, NAFLD and metabolic syndrome (MS) are two disorders sharing physio-pathological as well as clinical features, insulin resistance being the most important [33]. In patients with insulin resistance, an increase in visceral adipose tissue is observed, which is responsible for the release of different mediators from the adipose cells [34]. Furthermore, insulin resistance mediates the accumulation of intrahepatic fat through increased lipogenesis and lipolysis suppression [35], processes that further impair insulin signaling, increasing insulin resistance and thus promoting a vicious cycle [36].

Recently, a new concept was developed that led to the definition of MAFLD, a term that does not exclude alcohol consumption and other liver disorders associated with steatosis and better reflects the association of liver steatosis with an unhealthy lifestyle and metabolic disorders [37,38]. Therefore, MAFLD, unlike NAFLD, is no longer an exclusion diagnosis [30]. In a recent population study, it was reported that MAFLD had a prevalence of 25.9%, while NAFLD had a prevalence of 25.7% [30]. Furthermore, the authors observed that 89.2% of the subjects with liver steatosis met the diagnosis criteria for both MAFLD and NAFLD. The medical literature reports a prevalence of MAFLD up to 70%, in patients with T2DM, which raises up to 80% in the ones with metabolic syndrome [39]. In our study, the frequency of MAFLD was 49.5% in subjects with T2DM, a percentage comparable to the one reported by Samimi et al. (32.4%) in a cross-sectional study conducted in 2022 [23].

Another study published in 2022, analyzing NAFLD defined by liver biopsy in patients with metabolic syndrome, reported a total frequency of liver steatosis of 92.03%, ranging from 71.43% in patients with metabolic syndrome who had a normal weight to 91.84% in overweight patients, to 100% in patients with obesity [40]. Interestingly, in our research, anthropometric parameters were associated with MAFLD only in females.

In the last years, there was increasing interest in the search of different biomarkers that could not only non-invasively assess the liver function and establish a MAFLD diagnosis, but also be predictors for the unfavorable evolution of patients with MAFLD. In this regard, recently published studies demonstrated that the AIP is associated with NAFLD and MAFLD both in the general population [25,26] and in patients with T2DM [23]. AIP is a lipid indicator with predictive value for metabolic disorders (T2DM, dyslipidemia, hyperuricemia) as well as for cardiovascular diseases [27,41,42,43,44]. The study of Wang et al. [25] that enrolled 538 subjects with obesity demonstrated that AIP is a predictor of NAFLD in both males and females with obesity. In this study, the patients were divided into three groups depending on the AIP values, the high group (AIP > 0.21) having a 4.37-fold increase in NAFLD risk [25].

Another study conducted by Xie et al. in 7838 adults demonstrated that subjects with AIP in the higher quartile had a statistically higher risk of NAFLD (AUC = 0.810), especially females and at young ages [26].

Although some studies found a better association of AIP with NAFLD in subjects with obesity [45], the study of Dong et al. [27] conducted in a non-obese Asian population demonstrated a positive correlation between AIP and NAFLD in these subjects. In this study, the optimal cut-off value reported for Chinese subjects was 0.005, while for Japanese subjects, the optimal cut-off value was −0.22 [27]. Furthermore, using gender subgroup analysis, the authors noted that the association was closer in females compared to males, confirming previous results [44,46].

In a recent study including over 100,000 subjects, it was demonstrated that AIP is strongly correlated with NAFLD, with a cut-off value of 0.05 and an AUC of 0.82 [47]. Furthermore, the study also reported a sensibility of 78% and a specificity of 70% [47]. However, when comparing the results obtained by Liu et al. and the values reported by our study, we must take into account the fact that our study was conducted in patients with T2DM.

A recent meta-analysis including data from eight studies involving 81,178 subjects, which assessed the accuracy of AIP in predicting NAFLD, reported controversial results, the authors not finding statistically significant differences regarding the AIP levels between patients with NAFLD and the control subjects [48]. However, they observed significant difference in this parameter between males and females with NAFLD, reporting an AUC of 0.764 for males and of 0.733 for females [48]. In interpreting these results, we must not overlook the bias due to the heterogeneity between the studies, as was also noted by the authors [48]. To our knowledge, until now, only one other study [23] analyzed the potential of AIP as an independent predictor of MAFLD in subjects with T2DM. In this study, a cut-off value of 0.54 with 57.8% sensibility and 54.4% specificity was reported [23]. In our research, the AIP cut-off value that was predictive for MAFLD in patients with T2DM was 0.629, with a sensibility of 62.6% and a specificity of 37.6%. Furthermore, when discussing the predictive value of AIP for MAFLD, we should also take into account studies that demonstrated the utility of this biomarker in predicting atherosclerosis cardiovascular risk in different populations [45,49]. In patients with T2DM, the AIP was found both to be a risk factor for diabetes prognosis and to be associated with the risk of major adverse cardiovascular events, all-cause mortality, cardiovascular death, and nonfatal myocardial infarction [50]. Research prospective papers have also addressed the associations between NAFLD or MAFLD and cardiovascular risk, showing a similar significant increase in the risk of cardiovascular disease development in patients with NAFLD or MAFLD [51]. A recently published review of the literature highlighted the increased cardiovascular risk associated with NAFLD, showing that these patients have higher rates of coronary heart disease and cardiovascular events [52,53,54,55]. In addition, in patients with T2DM, a higher prevalence of coronary disease, cerebrovascular disease, and even peripheral arterial disease was observed when NAFLD was also present [52,56]. Further studies are needed in order to establish the prospective role of the AIP for the association MAFLD–cardiovascular disease.

A previous study conducted by Chen et al. [45] indicated that the TG/HDL ratio, an indicator from which the AIP was derived, is an independent predictor of NAFLD in non-obese people. In the present study, although the TG/HDL ratio presented a significantly higher median value in patients with T2DM and MAFLD, we could not prove the predictive value of this parameter for MAFLD in our study population. Intriguingly, increased values of serum TG were, however, predictive of MAFLD in patients with T2DM, but the cut-off value of 184 mg/dL had quite limited sensibility (53.5%) and specificity (33.7%).

Another lipid biomarker that was predictive of MAFLD in the patients with T2DM enrolled in this study was the non-HDL/HDL ratio, which presented a cut-off value of 3.9, associated with 62.6% sensitivity and 32.7% specificity. Recent studies demonstrated that this biomarker is a predictor of NAFLD in the general population. In the prospective study conducted by Wang et al. [57] in the Chinese population, it was concluded that the non-HDL/HDL ratio is a stronger predictor of NAFLD compared to the non-HDL level, the study reporting the optimal cut-off value of 2.3 in males and of 2.4 in females for the detection of NAFLD. These findings were also confirmed by Gao et al. [58], who recorded a cut-off value of this biomarker of 2.26, with 72.9% sensitivity and 58.2% specificity. Moreover, the utility of the non-HDL/HDL ratio in predicting NAFLD was also demonstrated in children and adolescents in whom the optimal cut-off values were 2.695 for girls and 2.475 for boys [59]. To our knowledge, our study is the first to assess the predictive value of the non-HDL/HDL ratio for MAFLD in patients with T2DM. However, future prospective studies are needed in order to introduce in the daily clinical practice these biomarkers for the evaluation of MAFLD in patients with T2DM.

In interpreting these results, it should also be considered the fact that, in our study group, a significant number of patients (89%) presented lipid disorders, with a prevalence slightly higher than that reported for hypercholesterolemia (69.1%) in a study conducted in the Romanian population aged 40–79 years [60]. In order to reduce bias and confounding factors, the analyses were adjusted for both the administration of a lipid-lowering therapy and the smoking status, which is a risk factor for lipid impairments [61]. Another issue that should be taken into account in the interpretation of the results is that a high percentage of our study population, both with and without MAFLD, presented obesity and a high waist circumference. In the interest of bias reduction and to be able to extrapolate the results to patients with T2DM who had a normal weight or were overweight, the results of the study were adjusted for both BMI and WC.

The biomarker with the highest AUROC (0.712) and the highest sensibility (74.7%) for the detection of MAFLD in patients with T2DM was HOMA-IR, with an optimal cut-off value of 2.01. This is not surprising, as insulin resistance is the underlying physiopathological mechanism of MAFLD, obesity, T2DM, dyslipidemia, metabolic syndrome [23]. A recent study performed by Masarone et al. in 135 patients with metabolic syndrome and 63 patients newly diagnosed with T2DM, who underwent liver biopsy after the clinical diagnosis of MAFLD, evidenced significantly higher HOMA-IR values in the patients with NASH compared to those with simple liver steatosis, suggesting that the severity of liver damage progressively increases with the increased presence of insulin resistance clinical manifestations [62]. Interestingly, in this study the frequency of NASH in patients with metabolic syndrome was 58.52%, while patients with T2DM presented NASH in a percentage of 98.6% [62]. These findings confirm previous results by Aller de la Fuente et al., who reported a frequency of NASH of 72.2% [63], and by Bian et al. [64], who reported a frequency of NASH of 96.1% in patients with T2DM. The results of these studies suggest that NASH is the main liver lesion encountered in patients with T2DM and MAFLD. In light of these observations, the results we obtained are very important for the future study of MAFLD in patients with T2DM, opening the perspective for anatomopathological studies that could assess the relationship between the biomarkers predictive of MAFLD and the severity and the progression of the liver disease. Furthermore, as it is the case for AIP, HOMA-IR is also a predictor of cardiovascular risk in patients with T2DM [65]. Therefore, future studies are necessary to establish the role of this biomarker in predicting the association of cardiovascular disorders with MAFLD in subjects with T2DM.

The results we presented must be also interpreted in the context of the study limitations. The main limitation of the study is its cross-sectional design, as it was an observational study, a fact that did not allow to define a cause–relationship effect. Therefore, we cannot draw pertinent conclusions regarding the predictive value of the biomarkers that were analyzed in relation to MAFLD and could only describe an association of these parameters with MAFLD at the time of observation. Furthermore, the relatively small number of subjects enrolled in the study represents another study limitation. The patients were randomly included in the study, which affected the mean age of the participants. Another important issue that must be taken into account is the fact that the study results cannot be applied to the general population, as the study was conducted in a limited geographical area.

## 5. Conclusions

MAFLD is an important comorbidity in patients with T2DM with a high prevalence, which was also confirmed in our study. It is important to find biomarkers that can identify patients with T2DM at risk for MAFLD progression, especially as studies have suggested that NASH is the most frequent manifestation of fatty liver in patients with T2DM. AIP and HOMA-IR, known markers of insulin resistance and cardiovascular disease, were good predictors of MAFLD in patients with T2DM.

## Figures and Tables

**Figure 1 diagnostics-12-02426-f001:**
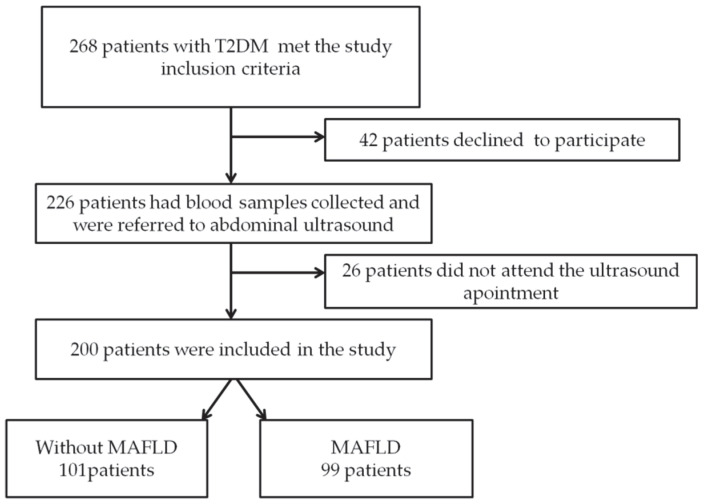
Study flow chart.

**Figure 2 diagnostics-12-02426-f002:**
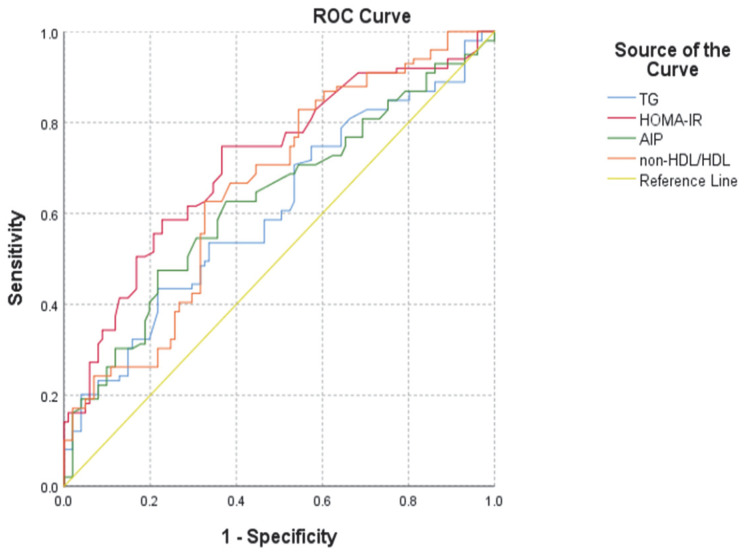
ROC curve for the biomarkers that were independently associated with MAFLD. AIP: atherogenic index of plasma; HDL: high-density lipoprotein; HOMA-IR: Homeostatic Model Assessment for Insulin Resistance.

**Table 1 diagnostics-12-02426-t001:** Anthropometric parameters associated with MAFLD.

Anthropometric Parameter		Group 1 (without MAFLD)	Group 2 (with MAFLD)	*p* Value
BMI (Kg/m^2^) *	Males	32 [6.05]	35.63 [5.55]	0.139
	Females	30.44 [3.66]	36.33 [10.43]	<0.001
WC (cm) *	Males	110 [14]	110 [17]	0.453
	Females	100 [19]	115 [15]	<0.001
WHR *	Males	1.03 [0.08]	1.04 [0.12]	0.985
	Females	0.97 [0.12]	0.97 [0.12]	0.471
WHtR *	Males	0.63 [0.06]	0.63 [0.09]	0.369
	Females	0.63 [0.13]	0.73 [0.12]	<0.001

* Variables with abnormal distribution, presented as median [IQR]. BMI: body mass index; IQR: interquartile range; WC: waist circumference; WHR: waist-to-hip ratio; WHtR: waist-to-height ratio.

**Table 2 diagnostics-12-02426-t002:** Biomarkers associated with MAFLD.

Biomarker	Group 1 (without MAFLD)	Group 2 (with MAFLD)	*p* Value
HbA1c (%) *	6.8 [1.86]	8.3 [2.8]	<0.001
Total cholesterol (mg/dL) *	186 [68.4]	199 [60]	<0.001
Triglycerides (mg/dL) *	153 [107.04]	192 [131]	0.009
HDL-cholesterol (mg/dL) *	44.4 [19]	40 [17.27]	0.013
LDL-cholesterol (mg/dL) *	105.79 [64.22]	120 [50.24]	0.017
AST (IU/L) *	22 [14.98]	23.34 [18.87]	0.127
ALT (IU/L) *	23 [27.03]	32 [26.51]	0.003
HOMA-IR *	1.96 [0.3]	2.3 [1.18]	<0.001
VAI *	5.93 [6.24]	8.85 [9.59]	0.01
AIP ^†^	0.53 ± 0.31	0.68 ± 0.36	0.002
non-HDL cholesterol (mg/dL) *	132 [74.1]	160 [52.04]	<0.001
Triglycerides/HDL ratio *	3.39 [3.43]	4.75 [6.33]	0.002
non-HDL/HDL ratio *	3.1 [2.74]	4.28 [2.3]	<0.001
ALT/AST ratio *	1.18 [0.58]	1.3 [0.71]	0.005

* Variables with abnormal distribution, presented as median [IQR]. ^†^ Variables with normal distribution, presented as mean ± SD. AIP: atherogenic index of plasma; ALT: alanine aminotransferase; AST: aspartate aminotransferase; HbA1c: glycated hemoglobin; HDL: high-density lipoprotein; HOMA-IR: Homeostatic Model Assessment for Insulin Resistance; LDL: low-density lipoprotein; VAI: visceral adiposity index.

**Table 3 diagnostics-12-02426-t003:** Lipid biomarkers associated with the smoking status.

Biomarker	Active Smoker	Former Smoker	Nonsmoker	*p* Value
Total cholesterol (mg/dL) *	196.22 [41]	198 [67.22]	195 [74.27]	0.806
Triglycerides (mg/dL) *	231 [140]	171 [267]	151 [127.06]	0.005
HDL-cholesterol (mg/dL) *	39.89 [17.43]	41 [22.21]	44.17 [17.83]	0.31
LDL-cholesterol (mg/dL) *	112 [59.68]	110.37 [43]	109 [55.74]	0.807
VAI *	9.56 [7.43]	5.72 [8.45]	6.87 [6.84]	0.028
AIP ^†^	0.76 ± 0.31	0.7 ± 0.34	0.55 ± 0.34	0.002
non-HDL cholesterol (mg/dL) *	156.33 [33.3]	161 [85.43]	151.13 [77.02]	0.647
Triglycerides/HDL ratio *	5.97 [5.31]	3.94 [8.92]	3.43 [4.07]	0.003
non-HDL/HDL ratio *	3.92 [1.9]	4.32 [4.12]	3.7 [2.66]	0.97

* Variables with abnormal distribution, presented as median [IQR]. ^†^ Variables with normal distribution, presented as mean ± SD. AIP: atherogenic index of plasma; ALT: alanine aminotransferase; AST: aspartate aminotransferase; HbA1c: glycated hemoglobin; HDL: high-density lipoprotein; LDL: low-density lipoprotein; VAI: visceral adiposity index.

**Table 4 diagnostics-12-02426-t004:** Backward stepwise regression for the biomarkers associated with MAFLD adjusted for gender, age, T2DM duration, lipid-lowering therapy, smoking status, BMI, and WC.

Biomarker	OR [95% CI]	*p*
Triglycerides	0.993 [0.988; 0.998]	0.010
AIP	42.139 [3.484; 509.634]	0.003
Non-HDL/HDL	0.576 [0.416; 0.796]	0.001
HOMA-IR	0.359 [0.213; 0.607]	<0.001

AIP: atherogenic index of plasma; HDL: high-density lipoprotein; HOMA-IR: Homeostatic Model Assessment for Insulin Resistance.

**Table 5 diagnostics-12-02426-t005:** AUROC and cut-off values of the biomarkers that were found to be independent predictors of MAFLD in patients with T2DM.

Biomarker	AUROC	*p*	Cut-Off Point	Sensibility	Specificity
Triglycerides	0.607	0.009	184	53.5%	33.7%
AIP	0.629	0.002	0.615	62.6%	37.6%
Non-HDL/HDL	0.659	<0.001	3.9	62.6%	32.7%
HOMA-IR	0.712	<0.001	2.01	74.7%	36.6%

AIP: atherogenic index of plasma; HDL: high-density lipoprotein; HOMA-IR: Homeostatic Model Assessment for Insulin Resistance.

## Data Availability

Not applicable.

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
