# Peer review of "A Study of Biomarkers Associated with Metabolic Dysfunction-Associated Fatty Liver Disease in Patients with Type 2 Diabetes"

_diagnostics, 2022, doi:10.3390/diagnostics12102426_

Round 1

Reviewer 1 Report

The current manuscript titled: "A study of biomarkers associated with metabolic dysfunction associated fatty liver disease in patients with type 2 diabetes" represents an important analysis of evolving field of Internal Medicine.

The title reflects the manuscript content and helps the reader navigate the article essence.

In my opinion, these are the adjustments which should be made to increase the value of your manuscript:

1.       In the manuscript title, please change all the first letters to capital.

2.       In abstract, add please abbreviations for “T2DM”.

3.       In Introduction chapter: add please more detailed information about NAFLD, including epidemiology and complications - to highlight the importance of this issue.

4.       There is not enough comparative information in the Discussions chapter with other similar studies. Please search for research data and add this information to this section.

5.       In Conclusions chapter, it is not clear practical relevance of this study: it appears that this study was conducted for further research only. Please formulate the conclusions more clearly and precisely with practical relevance clarification.

6.       The manuscript contains some punctuation errors, please revise the text.

Author Response

The current manuscript titled: "A study of biomarkers associated with metabolic dysfunction associated fatty liver disease in patients with type 2 diabetes" represents an important analysis of evolving field of Internal Medicine.

The title reflects the manuscript content and helps the reader navigate the article essence.

In my opinion, these are the adjustments which should be made to increase the value of your manuscript:

We thank the reviewer for the kind comments. We tried to address all the reviewer concerns as follows:

Point 1: In the manuscript title, please change all the first letters to capital.

Response 1: We thank the reviewer for the observation, we made the changes.

Point 2: In abstract, add please abbreviations for “T2DM”.

Response 2: We thank the reviewer for observing the error. We made the correction.

Point 3: In Introduction chapter: add please more detailed information about NAFLD, including epidemiology and complications - to highlight the importance of this issue.

Response 3: We absolutely agree, therefore we have added further information about NAFLD as requested by the reviewer.

Point 4: There is not enough comparative information in the Discussions chapter with other similar studies. Please search for research data and add this information to this section.

Response 4: Thank you for pointing this out. We discussed comparatively data from other studies, which were added to the References section of the manuscript.

Point 5: In Conclusions chapter, it is not clear practical relevance of this study: it appears that this study was conducted for further research only. Please formulate the conclusions more clearly and precisely with practical relevance clarification.

Response 5: We thank the reviewer for the observation and the thoughtful suggestions. We changed the Conclusion section of the manuscript accordingly.

Point 6: The manuscript contains some punctuation errors, please revise the text.

Response 6: Thank you for the comment. We have revisited orthography all through the manuscript.

Reviewer 2 Report

The manuscript is interesting, quite well written, even if with few elements of originality compared to the numerous papers on the subject.

This reviewer raises some issues that should be addressed by the authors.

1- The limitations section of the study is very short. It should be implemented. For example, a main limitation of the study is its cross-sectional observational design which does not allow to define a cause-effect relationship, and therefore a predictive value, between the biomarkers assessed and the MAFLD. This study can only describe an association at the time of observation. Another limitation is the relatively small number of patients enrolled in the study. These issues should be addressed by authors.

2- The population studied, both with and without MAFLD, was on average obese with a high waist circumference. These anthropometric characteristics of the population studied do not make the study results generalizable to a lean, and above all overweight, diabetic population, which accounts for a high percentage of type 2 diabetes.

3- The authors in the conclusions state that the findings of the study “… opening the perspective for anatomopathological studies (in type 2 diabetes) that could assess the relationship of the biomarkers predictive for MAFLD and the severity and the progression of the liver disease.” Recently it was observed by liver biopsy that steatohepatitis represents the sole feature of liver damage in type 2 diabetes. In other words, NAFLD/MAFLD generally presents as NASH in type 2 diabetic patients (PLoS One. 2017 Jun 1;12(6):e0178473. doi: 10.1371/journal.pone.0178473.). This interesting issue should be commented on in discussion.

4- NAFLD/MAFLD and insulin-resistance are bidirectionally correlated and, consequently, the development of pre-diabetes and diabetes is the most direct consequence at the extrahepatic level. Two very recent reviews explain in an updated and complete way the pathophysiological mechanisms that support this relationship (Antioxidants2021, 10(2), pp. 1–25, 270. doi: 10.3390/antiox10020270. - Processes2021, 9(1), pp. 1–18, 135. doi: 10.3390/pr9010135). Authors should comment and add the above references to the manuscript.

Author Response

The manuscript is interesting, quite well written, even if with few elements of originality compared to the numerous papers on the subject.

This reviewer raises some issues that should be addressed by the authors.

We thank the reviewer for the comments and suggestions. We tried to include all the reviewer’s recommendations in the new version of the manuscript, as follows:

Point 1: The limitations section of the study is very short. It should be implemented. For example, a main limitation of the study is its cross-sectional observational design which does not allow to define a cause-effect relationship, and therefore a predictive value, between the biomarkers assessed and the MAFLD. This study can only describe an association at the time of observation. Another limitation is the relatively small number of patients enrolled in the study. These issues should be addressed by authors.

Response 1: We thank the reviewer for pointing this out. We included all the suggestions in the Discussions section of the manuscript.

Point 2: The population studied, both with and without MAFLD, was on average obese with a high waist circumference. These anthropometric characteristics of the population studied do not make the study results generalizable to a lean, and above all overweight, diabetic population, which accounts for a high percentage of type 2 diabetes.

Response 2: We thank the reviewer for this comment. We addressed this issue in the Discussion section of the manuscript.

Point 3: The authors in the conclusions state that the findings of the study “… opening the perspective for anatomopathological studies (in type 2 diabetes) that could assess the relationship of the biomarkers predictive for MAFLD and the severity and the progression of the liver disease.” Recently it was observed by liver biopsy that steatohepatitis represents the sole feature of liver damage in type 2 diabetes. In other words, NAFLD/MAFLD generally presents as NASH in type 2 diabetic patients (PLoS One. 2017 Jun 1;12(6):e0178473. doi: 10.1371/journal.pone.0178473.). This interesting issue should be commented on in discussion.

Response 3: We thank the reviewer for the suggestion. We discussed the results of the suggested study and also modified the Conclusion section of the manuscript.

Point 4: NAFLD/MAFLD and insulin-resistance are bidirectionally correlated and, consequently, the development of pre-diabetes and diabetes is the most direct consequence at the extrahepatic level. Two very recent reviews explain in an updated and complete way the pathophysiological mechanisms that support this relationship (Antioxidants, 2021, 10(2), pp. 1–25, 270. doi: 10.3390/antiox10020270. - Processes, 2021, 9(1), pp. 1–18, 135. doi: 10.3390/pr9010135). Authors should comment and add the above references to the manuscript.

Response 4: Thank you for the suggestions. We discussed the requested references and added them to the reference list.

Round 2

Reviewer 1 Report

I agree with the changes made, which significantly improve the quality of the manuscript. I recommend this article for publication.

Reviewer 2 Report

No further comments